# Pre-Treatment of Swine Oviductal Epithelial Cells with Progesterone Increases the Sperm Fertilizing Ability in an IVF Model

**DOI:** 10.3390/ani12091191

**Published:** 2022-05-06

**Authors:** Costanza Cimini, Fadl Moussa, Angela Taraschi, Marina Ramal-Sanchez, Alessia Colosimo, Giulia Capacchietti, Samia Mokh, Luca Valbonetti, Israiel Tagaram, Nicola Bernabò, Barbara Barboni

**Affiliations:** 1Faculty of Biosciences and Technology for Food, Agriculture and Environment, University of Teramo, 64100 Teramo, Italy; ccimini@unite.it (C.C.); fmoussa@unite.it (F.M.); ataraschi@unite.it (A.T.); mramalsanchez@unite.it (M.R.-S.); acolosimo@unite.it (A.C.); gcapacchietti@unite.it (G.C.); lvalbonetti@unite.it (L.V.); itagaram@unite.it (I.T.); bbarboni@unite.it (B.B.); 2Doctoral School of Science, Technology Lebanese University, Beirut 1107, Lebanon; 3Istituto Sperimentale dell’Abruzzo e del Molise “G. Caporale”, 64100 Teramo, Italy; 4National Council for Scientific Research (CNRS), Lebanese Atomic Energy Commission (LAEC), Laboratory for Analysis of Organic Compound (LACO), Beirut 8281, Lebanon; samia.mokh@ul.edu.lb; 5Institute of Biochemistry and Cell Biology (CNRIBBC/EMMA/Infrafrontier/IMPC), National Research Council, 00015 Rome, Italy

**Keywords:** progesterone, in vitro fertilization, spermatozoa, sperm capacitation, oviduct, oviductal epithelial cells, *OVGP1*, SPP1, DMBT1, cytoskeleton

## Abstract

**Simple Summary:**

The involvement of the oviductal epithelial cells and progesterone hormone on the sperm fertilizing ability was investigated here by using an in vitro fertilization (IVF) swine model. By using different techniques, we demonstrated that oviductal epithelial cells (OECs) pre-treated with progesterone (P4) at 100 ng/mL represent the ideal environment for sperm capacitation, inducing a positive effect on the sperm acquisition of the fertilizing ability. Moreover, treatment of OECs with P4 modifies *OVGP1,*
*SPP1* and *DMBT1* gene transcription but does not interfere with important cytoskeleton structures or cell doubling time, thereby allowing the correct development and progression of important biochemical pathways and signal transduction processes.

**Abstract:**

Mammalian spermatozoa are infertile immediately after ejaculation and need to undergo a functional modification, called capacitation, in order to acquire their fertilizing ability. Since oviductal epithelial cells (SOECs) and progesterone (P4) are two major modulators of capacitation, here we investigated their impact on sperm functionality by using an IVF swine model. To that, we treated SOECs with P4 at 10, 100, and 1000 ng/mL before the coincubation with spermatozoa, thus finding that P4 at 100 ng/mL does not interfere with the cytoskeleton dynamics nor the cells’ doubling time, but it promotes the sperm capacitation by increasing the number of spermatozoa per polyspermic oocyte (*p* < 0.05). Moreover, we found that SOECs pre-treatment with P4 100 ng/mL is able to promote an increase in the sperm fertilizing ability, without needing the hormone addition at the time of fertilization. Our results are probably due to the downregulation in the expression of *OVGP1, SPP1* and *DMBT1* genes, confirming an increase in the dynamism of our system compared to the classic IVF protocols. The results obtained are intended to contribute to the development of more physiological and efficient IVF systems.

## 1. Introduction

Human infertility is estimated to affect approximately 48 million couples and 186 million individuals in the world [1,2], thus raising serious concerns in the last several decades. Currently, the efficacy of the assisted reproductive technology (ART) procedures is considered still considered to be far from optimal if compared with natural conception, in which the process of fertilization is subjected to the dynamics of the in vivo environment rather than to the static artificial conditions.

In vivo, the process of fertilization in mammals occurs within the oviduct, a complex structure that only a few spermatozoa are able to reach and where they strongly bind to the epithelial cells, forming a functional sperm reservoir [3,4]. During this storage, the interactions between the oviductal epithelial cells (OECs) and spermatozoa are thought to play an important role in sperm selection [5], sperm viability maintenance [6], and premature capacitation prevention [7]. After an indefinite period of time and around the time of ovulation, spermatozoa are released from the OECs to continue swimming towards the fertilization site, where they reach the egg [5]. The dynamic context created by the oviduct environment is characterized by the involvement of several molecules. Among these, progesterone (P4) plays an important role in spermatozoa. P4 is a steroid hormone produced in several organs and tissues, such as the placenta [8], the adrenal cortex, the ovary and the testis [9,10]. P4 secreted from the ovary arrives to the oviduct through a local countercurrent transfer [11], where it is able to act on the sperm cells. Among the multiple and wide-studied functions of P4 on spermatozoa it is worth to increasing the regulation of sperm hyperactivation [12] by increasing the intracellular concentration of calcium [Ca^2+^] mediated by CatSper channels (pH-dependent Ca^2+^ channel of the sperm flagellum) [13], the sperm chemotaxis towards the mature oocyte [14,15], the induction of acrosome reaction (AR) [16,17], and the promotion of capacitation in vitro [18,19]. Moreover, in the in vitro swine model, P4 has been observed to increase the sperm-binding capacity in to SOECs previously treated with E2 and P4, simulating estrus [20]. However, some important information is still missing regarding the effects caused by P4 directly on the oviductal cells that interact with the male gametes and that could also exert an indirect function on sperm cells.

The oviduct plays a key role as well also in the genetic and epigenetic reprogramming derived from the embryo-maternal interactions that take place during the early embryo development [21]. The significant increase in the prevalence of disorders generated to the embryo after ART procedures is worthy of note. For instance, the phenomenon of large offspring syndrome (LOS) in farm animals is characterized by macrosomia, macroglossia, omphalocele and abnormal organ growth [22], with a phenotypic similarity to the Beckwith-Wiedemann syndrome (BWS) in humans [22,23,24]. Many reports also showed the correlation between ART and Angelman syndrome (AS), a rare neurogenetic human syndrome characterized by severe mental retardation, the absence of speech, ataxia, seizures, an abnormal EEG pattern and hyperactivity [25,26]. The increasing prevalence of these pathologies presents an urgent need to improve ART protocols, progressing as much as possible towards the replication of the physiological conditions.

Several researchers around the globe are dedicated to the study of different approaches to improve IVF protocols, for instance by integrating oviductal fluid and uterine fluid in IVF [27,28] and embryo culture media [28], and with the introduction of 3-D oviductal epithelial cell culture systems to mimic the oviduct [29]. At present, the studies available to date are not focused on developing new strategies to increase the spermatozoa fertilizing ability.

Here, we propose the implementation of the sperm fertilizing ability using an IVF system that includes the somatic component (a monolayer of swine OECs, SOECs) and the addition of different P4 concentrations (10, 100, 1000 ng/mL [30,31,32]). We used a multiple-combinations strategy: on one hand, spermatozoa were capacitated in the presence of SOECs previously treated with P4; on the other hand, spermatozoa were capacitated both in the presence of SOECs and P4 at different concentrations. Thus, the final aim of this study was to evaluate the effects caused by P4 addition to both the somatic component and the gametes in an effort to approximate the IVF protocols to a more physiological condition.

## 2. Materials and Methods

### 2.1. Chemicals

Unless otherwise stated, all the chemicals were purchased from Sigma Aldrich (St. Louis, MO, USA).

### 2.2. SOECs Collection and Incubation

Swine oviducts at peri-ovulatory stages were collected from a local slaughterhouse and dissected. SOECs were gathered by scratching the whole oviduct (ampulla and isthmus tracts) following an established protocol [30,31]. Cells were pooled and then rinsed with a washing medium (TCM 199 supplemented with FBS 10%, penicillin/streptomycin 2% and amphotericin B 1%). SOECs were then cultured in 12-well plates in a medium containing TCM 199 supplemented with FBS 10% and penicillin/streptomycin 2% [29]. SOEC were treated with P4 at different concentrations depending on the experiment (10, 100, 100 ng/mL) and 17-β-estradiol (E2, 4 ng/mL), maintaining a control sample and incubated in a humidified atmosphere with 5% CO_2_ at 38.5 °C (Hera Cell Thermo Fisher Scientific, Monza, MB, Italy). The medium was renewed every 48 h.

### 2.3. Cell Population Doubling Time

To calculate the doubling time of SOECs, cells were cultured at a concentration of 1 × 10^6^ cells/mL, following the protocol already described and treated with the different P4 concentrations. After 48 h and seven days, cells were trypsinized and counted. The SOECs doubling time was calculated (Roth V. 2006 Doubling Time Computing, Available from: http://www.doubling-time.com/compute.php accessed on 1 March 2021) as:(1)$$DT=\frac{Duration\times log\left(2\right)}{log\left(FC\right)-log\left(IC\right)}$$
where:


*DT = doubling time*



*FC = final concentration*



*IC = initial concentration*


### 2.4. Phalloidin Staining

To assess the effect of progesterone treatment on actin polymerization, SOECs were cultured in 3.5 cm Petri dishes (38.5 °C, 5% CO_2_ and humidified atmosphere) and treated with the three different P4 concentrations. Once they reached 50% and 100% of confluence, SOECs were fixed overnight in paraformaldehyde 4%, then washed three times with PBS for 5 min each prior to the incubation with TRITC-conjugated phalloidin 488 (50 μg/mL in PBS, dilution 1:100) for 1 h. After three washes with PBS for 5 min, cell samples were mounted with Vectashield mounting medium (Vector H-1000) and subjected to microscopy analysis.

### 2.5. Tubulin Immunocytochemistry

SOECs were cultured with a cover glass, treated, and fixed as described above. After overnight fixation and washing, samples were permeabilized with 5% Triton X-100 in PBS for 30 min. SOECs were washed three times in PBS for 5 min and then incubated with BSA 5% + Normal Goat Serum (NGS) 5% in PBS for 30 min to block unspecific binding of the antibodies. Cells were then incubated overnight at 4 °C with monoclonal anti-mouse-α-tubulin antibody (SIGMA 025k4809) diluted 1:500 in PBS, containing 1% BSA. After washing, the bound antibody was detected using anti-mouse cy3 TRICT-conjugated diluted 1:500 in PBS containing 1% BSA incubated for 1 h at room temperature and followed by washing three times at 5 min intervals with 1% BSA—05% Tween20—PBS. Finally, the cover glasses were stained with DAPI 1:1000 and mounted in Vectashield mounting medium (Vector H-1000). SOECs samples incubated with PBS without the primary antibody were used as negative controls.

### 2.6. Confocal Analysis of SOECr

The acquisition was realized with a Nikon A1r laser confocal scanning microscope equipped with a Plan Apo λ 100X Oil objective and a Galvano detector with a pinhole size of 69 μm and a pixel size 0.04 µm. We used an averaged 2 mode in channels series as follows:

Channel 1: DAPI: λ_exc_ = 404 nm; λ_em_ = 450/50 nm

Channel 2: TRITC: λ_exc_ = 561.5 nm; λ_em_ = 595/50 nm

### 2.7. Preparation and Incubation of Sperm Samples

The preparation of boar semen samples was carried out following an already standardized protocol [33]. In brief, sperm samples were purchased from GENEETIC SRL, Modena, Italy. Before each experiment, sperm motility was visually estimated and only samples with a sperm motility >80% were considered for further analyses. To achieve capacitation, spermatozoa were incubated with confluent SOECs (after eight days of culture) in a capacitating medium TCM199 supplemented with 13.9 mM glucose, 1.25 mM sodium pyruvate, 2.25 mM calcium lactate and 1 mM of caffeine [34] at a final concentration of 1 × 10^7^ cells/mL and up to 1.5 h, at 38.5 °C in 5% CO_2_ humidified atmosphere (Heraeus, Hera Cell). The concentration of P4 100 ng/mL was added to the capacitation medium, preserving a control sample without the hormone.

### 2.8. In Vitro Fertilization

To obtain mature oocytes necessary for the IVF, ovaries from gilts were collected at a local slaughterhouse and transported to the laboratory within 1 h of slaughter maintaining a temperature of 25 °C. After washing the ovaries in a normal saline solution and bialcohol, the follicles of 4/5 mm diameter were selected on the basis of their translucent appearance, good vascularization and compactness of their granulosa layer and cumulus mass [35]. COCs (cumulus-oocyte complex) were collected by aspirating healthy selected follicles. The maturation process to the MII stage was obtained in vitro by culturing the COCs in four-well dishes containing α-MEM medium added with 10% FBS, 1% Penicillin/Streptomycin, 1% Ultraglutamine, hCG (5 UI/mL) and PMSG (5 UI/mL) for 44 h at 38.5 °C in a humidified atmosphere with 5% CO_2_ (Heraeus, Hera Cell) [35].

At the end of the maturation, the oocytes were denuded in DPBS with hyaluronidase on a warmed stage at 38.5 °C under a stereomicroscope. Only oocytes presenting the first polar body (MII stage) under the stereomicroscope were utilized for the IVF assay, using the same groups described above.

The day of the fertilization, swine spermatozoa were washed and incubated as previously described. Then, the capacitated spermatozoa at a final concentration of 1 × 10^6^ cells/mL were incubated with the oocytes in the presence of SOECs in a fertilization medium (capacitation medium supplemented with 10% FBS [35]). After 3 h, the potential fertilized oocytes were removed from the Petri dish, transferred into fresh medium and maintained in culture for at least 12 h. Subsequentially, the oocytes were fixed in Glutaraldehyde 0.5%, stained with Hoechst 33342 and observed with a fluorescence microscope.

As the experimental design (Figure 1) shows, IVF experiments were carried out in two steps: first, we perform the experiments using the combination of SOECs, spermatozoa and the three different P4 concentrations (10,100, 1000 ng/mL) to identify the most effective concentration (100 ng/mL) (Figure 1A), and then we performed the IVF using four different combinations, including SOECs not treated with P4 and sperm capacitation in the presence of P4, SOECs treated with P4 and sperm capacitation either with or without P4, and control conditions (Figure 1B). For both sets of experiments, the control group corresponding to spermatozoa capacitated in the presence of SOECs without pre-treatment with P4 nor P4 addition during capacitation.

IVF outcomes have been expressed as the fertilization rate (% of penetrated oocytes), the incidence of polyspermy (% of polyspermic oocytes), and the number of penetrating spermatozoa/polyspermic oocytes according to already published works [36,37]. We performed eight independent experiments, reaching a total number of 291 oocytes.

### 2.9. Gene Expression Analysis

Total RNA was extracted using the Total RNA Purification Kit (Cat. 17259, NORGEN Biotek Corp. Thorold, ON, Canada) according to the manufacturer’s instructions. Next, total RNA was evaluated and quantified using a Thermo Scientific NanoDrop 2000c UV-Vis spectrophotometer at 260 nm and stored at −80 °C until use in reverse transcription (RT). For the reverse transcription, 1 µg of total RNA from each sample was reverse transcribed into complementary DNA using an RT reaction with a Random Hexamers primer, (dNTP) mix and Tetro Reverse Transcriptase (Bioline, Luckenwalde, Germany) in a 20 µL reaction volume mixture according to the manufacturer’s instructions. cDNAs were diluted three times (60 µL final volume) and stored at −20 °C until used as template in an RT-qPCR mixture. Quantitative real-time PCR (qPCR) was carried out using primer sequences for *GAPDH*, *OVGP1*, *DMBT1* and *SPP1* genes that codify for glyceraldehyde 3-phosphate dehydrogenase and oviduct specific glycoprotein, deleted in malignant brain tu mors 1 and osteopontin proteins, respectively. The primers for *GAPDH* and *OVGP1* genes had been previously published [38]. The primers for *DMBT1* and *SPP1* genes were designed by using the Primer BLAST Tool (https://www.ncbi.nlm.nih.gov/tools/primer-blast/ 1 February 2021) and according to Thornton and Basu [39]. The primer sequences and their annealing temperatures are listed in Table 1. The specificity of the qPCR product was verified using the melting curve analysis program. The RT-qPCR reaction consisted of 13 µL master mix containing SensiFAST TM SYBR Lo-ROX kit (Bioline), primers and 2 µL of the diluted cDNA in a total volume of 15 µL, according to the manufacturer’s instructions. The cycling program was the two-step cycling protocol for 40 cycles (10 s at 95 °C for denaturation and 30 s at 60 °C for annealing/extension) with a 7500 Fast Real-time PCR System (Thermo Fisher Scientific, Monza, MB, Italy) followed by melt-profile analysis (7500 Software v2.3). For each qPCR analysis, each sample was performed in triplicate, and values were normalized to *GAPDH* endogenous reference gene (housekeeping gene). The relative gene expression was calculated by the comparative Ct (ΔΔCt) method and converted to the relative expression ratio (2^−ΔΔCt^).

### 2.10. Statistical Analysis

For statistical analysis, GraphPad Prism 6 Software (La Jolla, CA, USA) was used. Data were checked for normal distribution with a D’Agostino and Pearson normality test prior to perform the comparison with parametric or non-parametric tests, as required. In all cases the differences among groups were considered statistically significant when *p* < 0.05. To assess the effect of different treatments on IVF we carried out eight independent experiments using samples from different animals. An a priori power analysis was done to establish the number of oocytes with G*Power 3.1.9.7 software to get a final power of our analysis ≥95%. The results were expressed as the difference between the CTRL and treated samples, expressed as a percentage on the CTRL (Δ_IVF%_) [40]. To analyze the IVF data, Tukey’s and Dunnett’s multiple comparison tests were used.

To evaluate the effect of different treatments on gene expression, we carried out three independent experiments. The results of a D’Agostino and a Pearson normality test indicated a non-Gaussian distribution of the data and the controls are characterized by a variance = 0 (by definition the CRTL value is 1). Thus, we used the Kruskal Wallis test, and the data are expressed as median +/− range.

## 3. Results and Discussion

Fertilization is a complex process involving several essential components and an ensemble of connected processes. Among them, spermatozoa acquisition of the fertilizing ability through the process of capacitation is a key one, allowing them to interact with the mature oocytes within the oviduct. All these events are coordinated by the neuroendocrine axis through the dynamic changes induced by some steroid hormones such as P4, which modulates the oviductal environment and prepares the spermatozoa to fertilize the egg. The aim of the present work was to evaluate the action on the sperm fertilizing ability of two major modulators of capacitation, the oviductal epithelial cells and progesterone hormone, in an IVF model.

To achieve our goal, we performed two different sets of experiments:The identification of the optimal P4 concentration among three P4 concentrations (10, 100, 1000 ng/mL) [30,31] on morpho-functional parameters:Cell doubling timeActin organizationTubulin organization

The effects, in terms of sperm fertilizing ability in IVF assay, of SOECs exposure to P4 were calculated prior to their use as the environment for sperm capacitation.

Once the most favourable P4 concentration (100 ng/mL) was identified, we performed the comparison with further experimental conditions of interest (as shown in Figure 1B) to evaluate the sperm fertilizing ability, both with the SOECs treatment with or without P4 (100 ng/mL) and the addition or not of P4 (100 ng/mL) during capacitation.

### 3.1. P4 Supplementation Does Not Affect the Cells Growing Rate

We used different biological parameters to study the effects of P4 treatment on cell physiology. As with all the steroids, P4 is a lipophilic hormone, thus hampering the monitoring of its real bioavailable concentration regardless of the initial concentration of P4 with which SOECs are treated.

To explore whether the treatment of SOECs with P4 had effects on the cells’ growth and division, we evaluated the cells doubling time (DB) [41]. The cell doubling time is an important parameter used for the description of the cell development dynamics [42] and corresponds to the time it takes for a cell population to double in size [43]. It depends on different parameters: the time required for cell adhesion to the substrate; the cell adaptation to the artificial environment they are exposed to; and the time necessary for cell division [43]. In a cell culture, after the initial growth suppression a fast increase in the growth rate can be observed, followed by a constant exponential growth phase during the rest of the cell cycle [44]. Thus, with an exponential growth curve, the DB is an accurate measure of population cell growth [43]. We have calculated the DB of SOECs at two time points, 48 h and seven days. As shown in Figure 2, the doubling time was not affected by the presence of P4 at any of the three different concentrations (*p* > 0.05). Since this parameter is related with the ability of a cell culture to grow and adapt to the environmental conditions, these findings demonstrate that P4 does not exert a negative or toxic effect for the SOECs culture, independently on the concentration (within the range 10–1000 ng/mL).

### 3.2. P4 Supplementation Does Not Modify the SOECs Cytoskeleton

The cell cytoskeleton is composed of actin filaments, intermediate filaments and α- and β-tubulins that polymerize into microtubules. Due to the great importance of the cytoskeleton for the biological functions of cells such us migration, morphogenesis, cytokinesis, endocytosis, phagocytosis and response to extracellular stimuli [45], here we examined the effects of the SOECs pre-treatment with P4 directly on the actin filaments and tubulin patterns. Interestingly, either the F-actin organization and tubulin patterns remains unaffected by P4 treatment, as displayed respectively in Figure 3. 

Actins are a family of globular multi-functional proteins that polymerize to form filamentous structures. The most important physiological function of actin filaments is to produce force to be applied, for instance in cell migration and morphogenesis [46,47].

The other important protein examined here, tubulin, is a protein superfamily of globular proteins. α- and β-tubulins polymerize into microtubules, a major component of the eukaryotic cytoskeleton with an essential function in cell biology, such as mitosis, DNA segregation and cell division [48]. Tubulin and actin are involved in the dynamical arrangement of the cell cytoskeleton in response to intracellular (Ca^2+^, ATP and other second messengers) and extracellular (hormones, growth factors, ECM proteins) stimuli, and are able to control each other [49].

Abnormalities of these structures are associated with many pathological disorders. For instance, the altered expression of tubulin isotypes, alterations in tubulin post-translational modifications and changes in the expression of microtubule-associated proteins are associated with cancer [45,46,50]. Furthermore, the cytoskeleton alterations are involved in neurological [51], neurodevelopmental and neurodegenerative disorders [52]. For example, mutations in the *DCX* and *LIS1* genes that encode for microtubule-associated proteins are related to migration defects leading to the lack of development of brain folds and grooves [52].

Thus, from the results derived from the present work showing no alterations of these structures, we could affirm that P4 supplementation does not interfere with the development and progression of important biochemical pathways, thus not carrying a dysregulation of the signal transduction.

### 3.3. Sperm Capacitation on SOECs Previously Supplemented with P4 100 ng/mL Significantly Increases the IVF Outcomes

Based on the great importance of the oviduct and the neuroendocrine female axis in the fertilization process, we designed an IVF protocol using oviductal epithelial cells pre-treated with well-studied concentrations of P4 for the capacitation of the sperm cells [30,31,32].

Mammalian spermatozoa are not able to fertilize immediately after ejaculation, but the ability to fertilize the egg is only acquired after a series of changes that modify some of their biochemical, chemical, or physicochemical features within a process known as capacitation, which takes place during the transit of spermatozoa through the female genital tract [53,54]. This maternal environment operates a sperm selection as well, resulting in a very low percentage of sperm reaching the fertilization site. Thus, after swimming up the uterus and crossing the utero-tubal junction, the sperm arrive to the isthmus (the distal part of the oviduct), and release some messenger biomolecules that modulate the gene expression in epithelial cells before the physical contact [55].

Sperm cells bind preferentially to ciliated epithelial cells [56] through specific carbohydrate residues or the lectin-like proteins, thus forming the sperm reservoir [53,57]. Only motile, non-capacitated, acrosome-intact and normal chromatin structure spermatozoa are able to bind to the oviductal epithelium [58], extending their lifespan and delaying capacitation [59]. The mechanisms of sperm detachment are still partially unknown, but it is well-accepted that ovulation-associated signals induce the sperm release for the transit upper the oviduct tract (the ampulla) where fertilization occurs [5,60]. Numerous factors have been demonstrated to be involved in the release of spermatozoa, including ovarian steroid hormones [30,31,32,61], increased calcium levels [62], and anandamide [62], among others.

In our experiments, we first attempted to identify the most favorable P4 concentration to use as a supplement during SOECs growth prior to IVF. As it is evident from the IVF outcomes analysis (see Figure 4), the best results were obtained when performing capacitation on SOECs previously treated with P4 100 ng/mL, this concentration thus being considered to be the most effective one. It is possible to use the IVF assay to evaluate the sperm fertilizing ability in boar [37,63] due to the fact that the fertilization rates, the number of polyspermic oocytes and number of spermatozoa/polyspermic oocytes are related to the capacitation status and fertility of the semen [64,65]. Here, we used the IVF as a functionality test for spermatozoa to evaluate the capacitation status of sperm cells, instead that for producing embryos. In swine, in vitro maturated oocytes are partially unable to avoid the polyspermic fertilization [37,63], thus in specific conditions and in terms of sperm/oocytes ratio and length of coincubation it is possible to obtain polyspermic oocytes as a marker of their fertilizing ability. Multiple hypotheses arise that should be investigated to explain these results as the interaction between extracellular vesicles (EVs) and spermatozoa [66,67]. In a recent study, Asaadi and co-workers isolated EVs from oviductal fluid showing their presence and importance for fertilization and the embryo development in the oviductal environment [68]. However, further experiments should be done to shed some light into this interaction.

### 3.4. Pre-Treatment of SOECs with P4 but Not Sperm Capacitation Supplementation Improves IVF Outcomes

Once the most effective P4 concentration was identified, we performed a second set of IVF experiments to compare the IVF outcomes obtained when pre-treating the SOECs with P4 and the performance of the system when the hormone is added directly during sperm capacitation in the presence of SOECs (see the experimental design of Figure 1B).

The new data (illustrated in Figure 5) confirmed the results previously obtained: spermatozoa capacitated with SOEC pre-treated with P4 represented the most favorable condition with respect to the groups where P4 (100 ng/mL) was added during capacitation. From the data displayed in Figure 5, it emerges that the pre-treatment of SOECs showed the best performance due to the greater values obtained in the percentage of polyspermic oocytes that allow the inference of the sperm’s fertilizing ability [37,63,64,65].

### 3.5. P4 Induces a Downregulation of the Expression of OVGP1, SPP1 and DMBT1 Genes

Once the most effective P4 concentration for IVF protocol was identified, we evaluated the effect of P4 (100 ng/mL) treatment on gene expression of SOECs, using as a positive control SOECs treated with 17-β-estradiol (E2, 4 ng/mL). We performed and validated a RT-PCR robust system in which the mRNA expression of the three following genes was evaluated: oviductin (*OVGP1*), osteopontin (*SPP1*), or deleted in malignant brain tumors 1 (*DMBT1*).

These three genes are critical in sperm capacitation and fertilization in vitro, since previous studies have shown that their increased expression has been associated with the increase of the rate of monospermic fertilization in swine, through different mechanisms, thus resulting in an increase of normal fertilization in vitro [69]. Other specific effects of each gene on reproduction are reported below.

The *OVGP1* gene encodes for the oviduct specific glycoprotein (OVGP1, also called oviductin) [70], which is produced by the non-ciliated epithelial cells of the oviduct and secreted into the lumen [70,71]. OVGP1 has multiple important roles deeply related to fertilization [71,72] conferring by association a higher resistance to the zona pellucida (ZP) before fertilization [69,71]. Moreover, some positive effects have been conferred to the protein during the process of sperm capacitation in terms of viability, motility and fertilization ability, [72], sperm-egg binding and early embryo development [73].

The *SPP1* gene codifies for osteopontin (OPN), a phosphoprotein that was first found in the mineralized matrix of bovine bones [69,74] and subsequentially identified in various tissues, including male and female reproductive tracts [74,75]. OPN is involved in immune regulation, matrix communication, cell adhesion, cell migration and bone homeostasis [76]. In the reproductive field, it may play a role in sperm-egg interaction. Indeed, OPN may be able to bind the integrins of the oocytes by its GRGDS sequence, reducing the likelihood of sperm binding [69,74].

Finally, the *DMBT1* gene encodes for deleted in malignant brain tumors 1 protein (DMBT1). DMBT1 is a multifunctional glycoprotein present in epithelial cells, mucosal surfaces and the skin [77]. It has been identified as playing several roles in the innate immune defense system against pathogens, homeostasis, inflammation and tumor suppression [77,78,79]. In reproductive organs, DMBT1 has been detected in the oviduct of *sus scrofa* and human females, and is related to the ability of sperm binding and probably involved in the homeostasis of the epithelium covering the female tract [80,81,82].

Previous studies have shown that steroid hormones might affect the gene expression of the oviductal epithelium both in vitro and ex vivo [38,83,84]. Our results show a consistent downregulation of *OVGP1, SPP1* and *DMBT1* genes in SOECs pre-treated with P4, as well as a significant higher expression of these genes in SOECs treated with E2 (Figure 6) in comparison to the control. These data confirm that *OVGP1* and *DMBT1* gene expression is estrogen-dependent [69]. Moreover, our findings demonstrated that P4 administration downregulated the expression of the *SPP1* gene, while estradiol also increased the expression of this gene (Figure 6B). Previous works reported a regulation of *SPP1* gene expression in a species-specific manner. For instance, in bovine species the gene expression was not correlated to the estrous cycle phases [75], while in other small mammals as the mouse the expression was highest in estrus and lowest in diestrus, indicating that it may be regulated by E2 [85].

The higher percentage of polyspermic oocytes observed as a consequence of the administration of P4 100 ng/mL (Figure 5) is fully concordant with the decrease in mRNA expression of *OVGP1*, *SPP1* and *DMBT1*, in the same experimental setting.

In line with these results, a previous study demonstrated that increased levels of serum progesterone induce an evident inhibitory effect on *OVGP1* mRNA expression in oviductal mucosal tissue from fertile women [86].

Quantitative RT-PCR data show changes in the bioactivity of SOECs treated with P4 with respect to the control group. This confirmed an increase in the dynamism of our system compared to the classic IVF protocols, while partly explaining the increased sperm fertilizing ability. Further proteomic analyzes are needed to evaluate the real presence of the proteins of interest.

## 4. Conclusions

In conclusion, despite its limitations, our study shed light into the functional link between sperm capacitation and two important modulators of sperm physiology, SOEC and P4. It clearly demonstrates that the incubation of male gametes in the presence of SOECs pre-treated with P4 (100 ng/mL) enhances their fertilizing ability.

These findings could be very important for two reasons. First, they suggest the existence of new mechanisms that can drive capacitation, with the implication in the understanding of the physiology and the pathology of that process. They could then set the basis for the design of innovative approaches to promote capacitation in a dynamic in vitro environment, not only for human ARTs but also for farm animals’ embryo production.

Extending our understanding on the involvement of the main modulators of sperm capacitation in the oviductal environment as with the oviductal epithelial cells and progesterone remains essential in order to design new strategies not only for human ARTs but also for farm animals’ embryo production.

## Figures and Tables

**Figure 1 animals-12-01191-f001:**
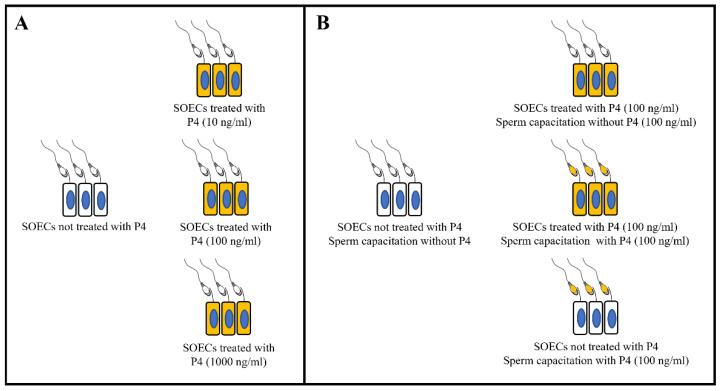
Experimental design for IVF. (**A**) Experimental groups to identify the most effective P4 concentration. (**B**) Experimental groups using the most effective P4 concentration and sperm capacitation with and without P4 (100 ng/mL) in the presence of SOECs previously treated or not with P4.

**Figure 2 animals-12-01191-f002:**
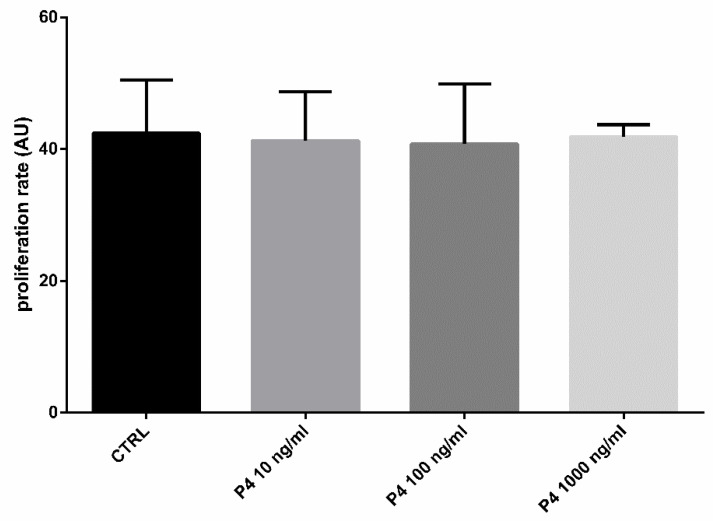
Effect of the different P4 concentrations on SOECs doubling time. The histograms show a normal proliferation rate of the samples treated with different concentrations of P4 (10, 100 and 1000 ng/mL), similar to the control (CTRL) group (*p* > 0.05). Three independent experiments were performed.

**Figure 3 animals-12-01191-f003:**
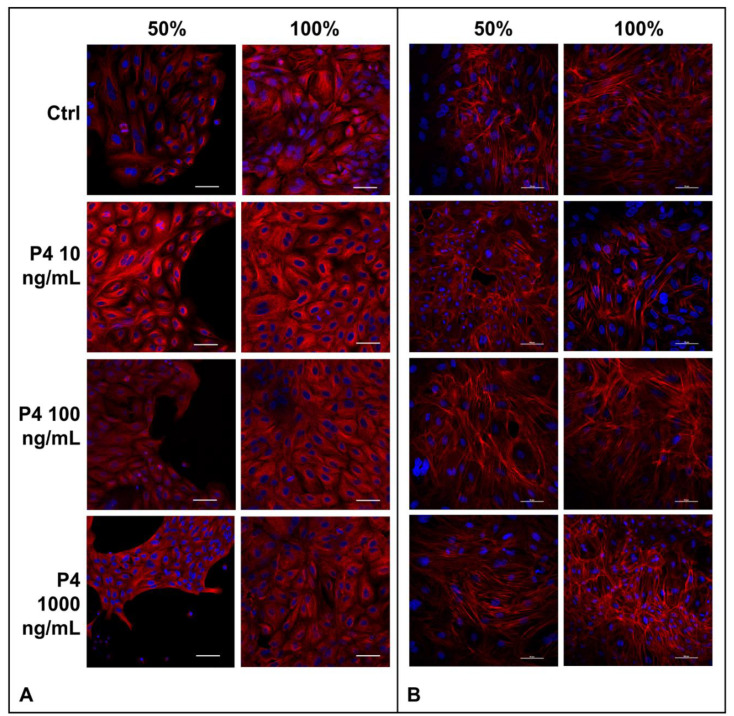
(**A**) Microtubules organization of SOECs. Immunofluorescence analysis of α-tubulin of SOECs after P4 treatment at different confluence time points (50% e 100%) and control condition (CTRL). Monoclonal anti-mouse-α-tubulin antibody and secondary antibody anti-mouse cy3 TRICT-conjugated (red) were used to detect α-tubulin; DAPI was used to stain the nuclei (blue). Three independent experiments were performed. (**B**) Actin filaments of SOECs. Phalloidin staining was used to study the F-actin of SOECs after P4 treatments at different confluence time points (50% e 100%) and control condition (CTRL). DAPI (blue, nuclei), TRITC-conjugated phalloidin (red, F-actin). Three independent experiments were performed.

**Figure 4 animals-12-01191-f004:**
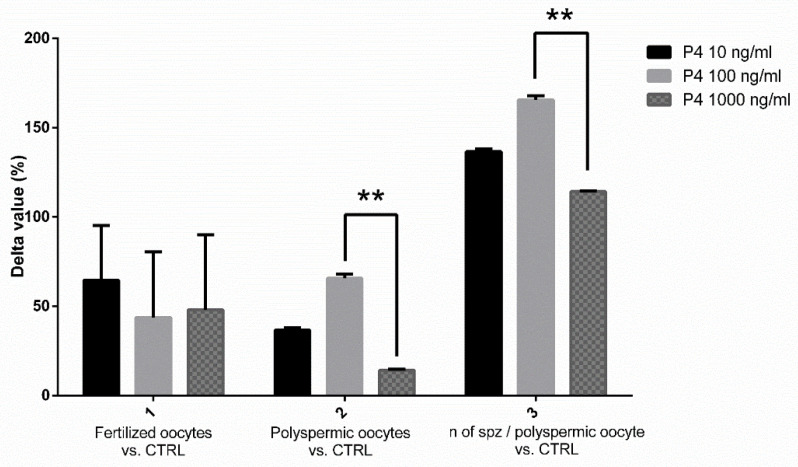
Graphical representation of the effects of P4 at different concentrations. Results are ex-pressed as the difference in percentage (Δ IVF%) of the fertilization rates, the number of polyspermic oocytes and the number of spermatozoa per polyspermic oocyte, comparing the groups of spermatozoa capacitated in the presence of SOEC treated with P4 (at 10,100 and 1000 ng/mL) to the control group without P4. The best IVF results were obtained when SOECs were treated with P4 at 100 ng/mL, due to the higher values obtained with this concentration in terms of difference of percentage in the number of polyspermic oocytes and in the number of spermatozoa/polyspermic oocyte. The data are presented as mean of four independent experiments. Data were analyzed using a Tukey’s test ** *p* < 0.01.

**Figure 5 animals-12-01191-f005:**
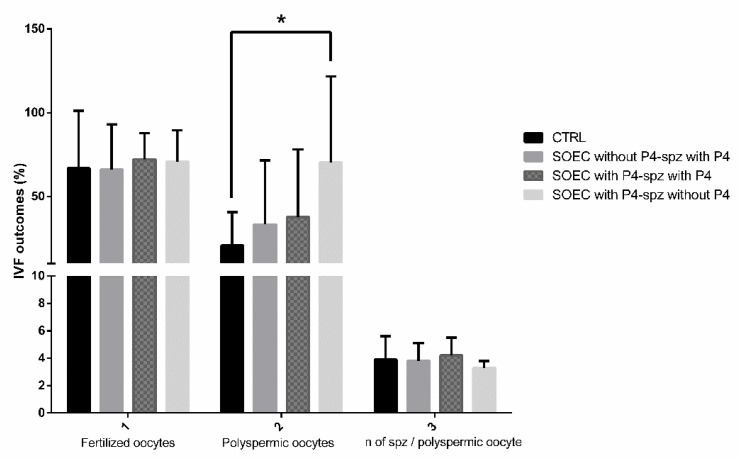
IVF outcomes: comparison between pre-treatment of SOECs with P4 and addition of P4 during sperm capacitation. The graph shows the effects of co-incubation of sperm during capacitation with P4 (100 ng/mL) and SOECs pre-treated with P4 (100 ng/mL) on IVF outcomes. By analyzing the graph, it arises that the best IVF outcomes are obtained when SOECs are pre-treated with P4 at 100 ng/mL and sperm capacitation is performed without the addition of P4. The data are presented as mean of four independent experiments. Data were analyzed using a Dunnett’s test. * *p* < 0.05 versus control.

**Figure 6 animals-12-01191-f006:**
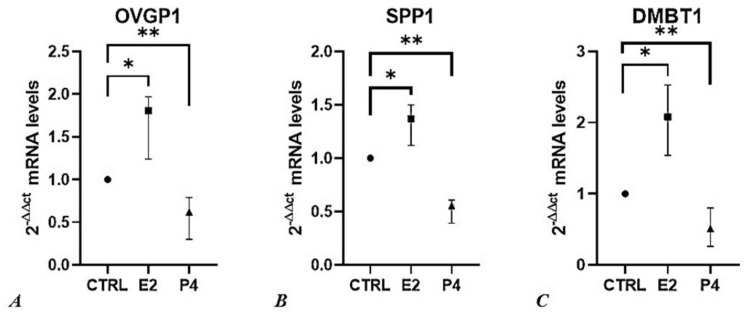
The effect of progesterone and estradiol on relative gene expression (2^−ΔΔCt^) of (**A**) *OVGP1*, (**B**) *SPP1*, and (**C**) *DMBT1*. Results are represented as median value with 95 % confidence interval. Data were analyzed using the Kruskal-Wallis test. For each qPCR analysis, each sample was performed in triplicate, and values were normalized to the *GAPDH* endogenous reference gene. P4, Progesterone; E2, Estradiol; ** = *p* < 0.001, * = *p* < 0.01.

**Table 1 animals-12-01191-t001:** Primer sequences and corresponding annealing temperatures for RT-qPCR analysis.

Gene Symbol	Primer Sequence (5′–3′)	Fragment Size (bp)	Anneling Temperature (°C)	References
*GAPDH*	Forward: ATTCCACCCACGGCAAGTTC			
	Reverse: AAGGGGCAGAGATGATGACC	225	60	[38]
*OVGP1*	Forward: TACTTGAAGAGCTCCTGCTTGCCT			[38]
	Reverse:TCTTCCCAGAAGGCGCACATCATA	134	60	
*DMBT1*	Forward: GAAATAGAGGTGAACTCCGGCTA-	165	60	Primer-BLAST
	Reverse: TGTGAATATCTGGCTGGTGTGAT			
*SPP1*	Forward: GCCCTTCCAGTTAAACAGACTAAT			
	Reverse: AGGGTCTCTTGTTTGAAGTCGT	176	60	Primer-BLAST

## Data Availability

The data presented in this study are available in the article.

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
