# Peer review of "Pre-Treatment of Swine Oviductal Epithelial Cells with Progesterone Increases the Sperm Fertilizing Ability in an IVF Model"

_animals, 2022, doi:10.3390/ani12091191_

Round 1
Reviewer 1 Report
In this manuscript entitled “Pre-treatment of swine oviductal epithelial cells with progesterone increases the sperm fertilizing ability in an IVF model” authors show that SOECs treatment with P4 induced better sperm capacitation in comparison with control. The manuscript is well written, nevertheless, I have some doubts and comments that authors should address before to be recommended to publish.
For instance:
Line 251: Please change “of the two main modulators” by “of two of the main modulators”
Please described in each figure legend how many times a set of experiments was repeated and described the statistic used.
Figure 1B, first row: Please check “Sperm capacitation either with P4”. Did you mean “Sperm capacitation without P4”
Figure 3. Authors stated that there are not differences between the different concentration of P4 used and control. Nevertheless, it seems that there are differences between all the treatment in figure 3A looking to the images of 100% confluence. For instance, in P4-100ng/mL the is no signal for tubulin. In addition, the figures have very low quality. For instance: control and p4 10ng/mL are too flurry when 100% is achieved. Same issue is observed in Figure 3B. Please review the figure and be sure that these are the better that you can show. It is expected that confocal microscopy obtains better quality images.
Although the reviewer appreciates the innovative way to present result, I recommend to authors to change the Figure 4 and 5 to the classic histogram, thus, the reader will appreciate much better the real values and statistic. Moreover, as the authors mention in the text, with the classical histogram graph the reader will appreciate the standard errors of the experiments.
Line 364-372. Authors claims that P4-100ng/mL is the better. Why this concentration is better? it seems that the percentage of fertilized oocytes is at the same level that p1000 and lower than p 10. Please could you explain better this statement?
Line 401-405-In figure legend it should not be described results. Please correct this issue.
Line407-408. Is should be included in the figure legend.
Line-419-421.This is the contrary to what the authors claims, SOECs treated with P4 induces better sperm capacitation, nevertheless P4 decrease the expression of OVCP1. Why there is not statistic in the figure. It seems that between control and P4 there is a significant difference. Please, check this issue.
Figure 6. Please insert the statistical analysis.
Authors should hypothesize about how the treatment of SOECs with P4 induces better sperm capacitation. It might be related with the quantity and/or quality of EVs produced by SOECs during the treatment? I encourage authors to follow this line of research.
Author Response
The Authors thank the Editor and Referees for their careful and rapid review of our manuscript. All the modifications suggested by the Referees have been appreciated and followed, and we consider that the manuscript has considerably improved now. Please find enclosed a clean version of the revised manuscript (Animals_cleanversion) and a file with all the highlighted modifications (Animals_highlighted). As requested by the Referees, the figures regarding the IVF results have been modified and better explained. The Conclusion section has been extensively revised.
We really hope that you could reconsider our new version of the manuscript suitable for publication in Animals. We remain at your disposal in case you need any other clarification or additional information.
Looking forward to hearing from you, we send you our best regards.
Reviewers'comments:
Reviewer #1: In this manuscript entitled “Pre-treatment of swine oviductal epithelial cells with progesterone increases the sperm fertilizing ability in an IVF model” authors show that SOECs treatment with P4 induced better sperm capacitation in comparison with control. The manuscript is well written, nevertheless, I have some doubts and comments that authors should address before to be recommended to publish.
Line 251: Please change “of the two main modulators” by “of two of the main modulators”
Reply: We sincerely thank Reviewer 1 for his/her careful review, nice comments, and constructive suggestions. We have modified the sentence. Please see L254.
Please described in each figure legend how many times a set of experiments was repeated and described the statistic used.
Reply: Thank you for your comment. We have added the number of experiments and the statistical analysis used in the figure legends.
Figure 1B, first row: Please check “Sperm capacitation either with P4”. Did you mean “Sperm capacitation without P4”
Reply: We apologize for this mistake. We have corrected this sentence.
Figure 3. Authors stated that there are not differences between the different concentration of P4 used and control. Nevertheless, it seems that there are differences between all the treatment in figure 3A looking to the images of 100% confluence. For instance, in P4-100ng/mL the is no signal for tubulin. In addition, the figures have very low quality. For instance: control and p4 10ng/mL are too flurry when 100% is achieved. Same issue is observed in Figure 3B. Please review the figure and be sure that these are the better that you can show. It is expected that confocal microscopy obtains better quality images.
Reply: Thank you for your suggestion. We have changed the figure 3 with better quality images and we hope you could find them suitable now.
Although the reviewer appreciates the innovative way to present result, I recommend to authors to change the Figure 4 and 5 to the classic histogram, thus, the reader will appreciate much better the real values and statistic. Moreover, as the authors mention in the text, with the classical histogram graph the reader will appreciate the standard errors of the experiments.
Reply: We agree. Accordingly, the reviewers’ suggestion we changed the graph typology and the statistical analysis. We added the graphical notation for statistically significative differences.
Line 364-372. Authors claims that P4-100ng/mL is the better. Why this concentration is better? it seems that the percentage of fertilized oocytes is at the same level that p1000 and lower than p 10. Please could you explain better this statement?
Reply: Thank you for your comment. Here, we used the IVF as a functional test for spermatozoa, without the aim of obtaining embryos. It means that our work was set up to evaluate the capacitation status of sperm cells in a specific context: after the SOECs treatment with P4. As it has been reported in literature, in swine in vitro maturated oocytes are partially unable to avoid the polyspermic fertilization, thus in specific conditions in term of sperm/oocytes ratio and length of coincubation, it is possible to obtain polyspermic oocytes as a marker of the fertilizing ability. Thus, the best performance, due to the greater values obtained in the percentage of polyspermic oocytes and number of spermatozoa/polyspermic oocytes that allow to infer the sperm fertilizing ability, was obtained when using P4 at 100 ng/mL.
Line 401-405-In figure legend it should not be described results. Please correct this issue.
Reply: Thank you, we have corrected this mistake.
Line407-408. Is should be included in the figure legend.
Reply: Thank you, we agree, this mistake was corrected.
Line-419-421.This is the contrary to what the authors claims, SOECs treated with P4 induces better sperm capacitation, nevertheless P4 decrease the expression of OVCP1. Why there is not statistic in the figure. It seems that between control and P4 there is a significant difference. Please, check this issue.
Reply: We apologize for the misunderstanding, eventually found in the initial version of our manuscript.
The purpose of the present study was to analyze the effect of progesterone (P4) administration on sperm fertilizing ability, in terms of % of fertilized oocytes, number of spermatozoa/polyspermic oocyte and % of polyspermic oocytes.
Previous studies have shown that an increased expression of some oviductal proteins, such as oviductin (OVGP1), osteopontin (SPP1) or deleted in malignant brain tumours 1 (DMBT1), reduces the rate of polyspermy in the pig, through different mechanisms, thus resulting in an increase of normal fertilization in vitro (see Coy & Yanagimachi, 2015).
In this study we have shown that the administration of progesterone (P4) (100ng/ml) increases the sperm fertilizing ability, due to the higher percentage of polyspermic oocytes (fig. 5).
In this regard, the increased incidence of polyspermy observed in our IVF swine model is fully concordant with the slight decrease in mRNA expression of OVGP1, SPP1 and DMBT1 (fig.6), following the P4 (100ng/ml) administration.
In concordance with these results, a previous study has demonstrated that increased levels of serum progesterone induce an evident inhibitory effect on OVGP1 mRNA expression, in oviductal mucosal tissue from fertile women (Regulation of human oviductin mRNA expression in vivo. C Briton-Jones, I H Lok, P M Yuen, T T Chiu, L P Cheung, C Haines. Fertil Steril. 2001 May;75(5):942-6. doi: 10.1016/s0015-0282(01)01696-x).
Nevertheless, as mentioned in the text, we are aware that protein analysis is needed to confirm the gene expression results.
Figure 6. Please insert the statistical analysis.
Reply: We apologize for this mistake. The statistical analysis has been added to Figure 6 (figure and legend).
Authors should hypothesize about how the treatment of SOECs with P4 induces better sperm capacitation. It might be related with the quantity and/or quality of EVs produced by SOECs during the treatment? I encourage authors to follow this line of research.
Reply: Thank you for this interesting suggestion, that for sure we will consider in further works. Actually, we have multiple hypothesis that should be studied. For instance, P4 could induce an increase in the expression of proteins and receptors on SOECs, that could somehow improve the sperm capacitation during the sperm attachment to the cells (physical contact). Surely, the release of EVs is another hypothesis (alone or in combination with the first one), which not include a physical contact between the two elements. However, further experiments should be done to shed some light into this interaction.
References:
Asaadi, A.; Dolatabad, N.A.; Atashi, H.; Raes, A.; Damme, P. Van; Hoelker, M.; Hendrix, A.; Pascottini, O.B.; Soom, A. Van; Kafi, M.; et al. Extracellular vesicles from follicular and ampullary fluid isolated by density gradient ultracentrifugation improve bovine embryo development and quality. Int. J. Mol. Sci. 2021, 22, 1–16, doi:10.3390/ijms22020578.
Franchi, A.; Moreno-Irusta, A.; Domínguez, E.M.; Adre, A.J.; Giojalas, L.C. Extracellular vesicles from oviductal isthmus and ampulla stimulate the induced acrosome reaction and signaling events associated with capacitation in bovine spermatozoa. J. Cell. Biochem. 2019, 1–12, doi:10.1002/jcb.29522.
Almiñana, C.; Tsikis, G.; Labas, V.; Uzbekov, R.; Silveira, J.C.; Bauersachs, S.; Mermillod, P. Deciphering the oviductal extracellular vesicles content across the estrous cycle : implications for the gametes-oviduct interactions and the environment of the potential embryo. 2018, 1–27
Reviewer 2 Report
The abstract is not clear; the authors must clarify the experimental groups.
The introduction does not comment anything on the effect of progesterone on POEC or spermatozoa.
The authors do not justify the gene selectionGAPDH, OVGP1, DMBT1, SPP1. Why these and not others?
The paragraph from line 41 to line 50 is unnecessary for this paper. It does not contribute with anything.
Line 142, what did you use, slides or dish?
How did the authors prepare the spermatozoa for IVF? “spermatozoa were incubated with confluent SOECs” . Did the authors use spermatozoa not attached to POEC? Please explain this step better. The spermatozoa that do not bound to POEC are of “bad quality”, did you use those? On the other hand, if the spermatozoa to perform the IVF are bound (they are the “best”), then how do you calculate the concentration? Please, be clearer on how you managed this experiment.
In material and methods, it says that 291 oocytes were used in total, but was it per replicate or in all 8 experiments? In the latter case, it would have been 36 oocytes per experiment, and that number is very low when working with porcine IVF.
Figures 4 and 5 are not adequate to visualize the results or the differences between groups. They do not allow certain calculations to be made, such as the average number of sperm, number of oocytes per replicate...
Line from 302 to 319: This paragraph is not a discussion of your results, it is a protein review (actine/tubuline).
In line 411 it is the first time that the use of estrogens is mentioned. The authors show results of estrogens on POEC, but do not explain how they do the experiment in material and methods.
The conclusion is not clear if progesterone is good for POEC, or if POEC treated with P4 are better for preparing sperm, or that only p4 on POEC culture does not affect POEC but does affect capacitation. (already known fact). In any case, there is not a single study on sperm capacitation (motility, protein phosphorylation...).
Author Response
The Authors thank the Editor and Referees for their careful and rapid review of our manuscript. All the modifications suggested by the Referees have been appreciated and followed, and we consider that the manuscript has considerably improved now. Please find enclosed a clean version of the revised manuscript (Animals_cleanversion) and a file with all the highlighted modifications (Animals_highlighted). As requested by the Referees, the figures regarding the IVF results have been modified and better explained. The Conclusion section has been extensively revised.
We really hope that you could reconsider our new version of the manuscript suitable for publication in Animals. We remain at your disposal in case you need any other clarification or additional information.
Looking forward to hearing from you, we send you our best regards.
Reviewer #2The abstract is not clear; the authors must clarify the experimental groups.
Reply: We thank Reviewer 2 for his/her attentive reading, positive appreciation, and pertinent questions. We have corrected and clarified the experimental groups in the abstract and we hope it is clear now.
The introduction does not comment anything on the effect of progesterone on POEC or spermatozoa.
Reply: Thanks for your comment. We have included a reference regarding the effects of P4 on sperm binding to oviductal cells in vitro. Moreover, P4 effects on spermatozoa are described in L64-65
Reference:
Chen, S.; Einspanier, R.; Schoen, J. In vitro mimicking of estrous cycle stages in porcine oviduct epithelium cells: Estradiol and progesterone regulate differentiation, gene expression, and cellular function. Biol. Reprod. 2013, 89, 1–12, doi:10.1095/biolreprod.113.108829.
The authors do not justify the gene selection GAPDH, OVGP1, DMBT1, SPP1. Why these and not others?
Reply: We chose the three target genes (OVGP1, DMBT1, SPP1) because they are critical in sperm capacitation and fertilization in vitro, since their increased expression has been associated with the increase of the rate of monospermic fertilization in swine, besides other effects. To point out their specific role in decreasing the risk of polyspermy in the pig, we have added a sentence in the revised text (L435-441)
In addition, since two of them (OVGP1 and DMBT1), have been shown to be estrogen-dependent glycoproteins in the pig, we used estradiol treatment as a further control of our experimental conditions.
The paragraph from line 41 to line 50 is unnecessary for this paper. It does not contribute with anything.
Reply: Thanks for your comment, we have modified the Introduction section, L41-46.
Line 142, what did you use, slides or dish?
Reply: We apologize for this typing mistake. We used cover glass, see L140.
How did the authors prepare the spermatozoa for IVF? “spermatozoa were incubated with confluent SOECs”. Did the authors use spermatozoa not attached to POEC? Please explain this step better. The spermatozoa that do not bound to POEC are of “bad quality”, did you use those? On the other hand, if the spermatozoa to perform the IVF are bound (they are the “best”), then how do you calculate the concentration? Please, be clearer on how you managed this experiment.
Reply: To our knowledge, and thanks to the several experiments performed by our group and others, we consider the spermatozoa as dynamic cells, also within the oviduct tract and during their attachment. In fact, it is well-known that spermatozoa are able to attach and detach from the epithelium during their way towards the oocyte, both within the isthmus and the ampulla tracts. By an unknown mechanism and after and indefinite period of time, spermatozoa detach from the epithelium and continue their trip. In this work, we used a pool of SOECs using segments from the isthmus and the ampulla, and let the spermatozoa capacitate in the presence of the cells, trying to approach the in vitro system to a more physiological strategy. For this reason, we cannot be sure of which spermatozoa are of “bad quality” or the “best”, since it is probably that those spermatozoa swimming (thus, not attached) after 90 min of co-incubation with the oviductal epithelial cells had acquired a capacitating state that allow them to fertilize the oocyte. The concentration is then calculated to use a number of spermatozoa of 10^6, although we agree with this R that we cannot be sure of the real number. Due also to this fact, the experiment was repeated several times (8).
References:
Romero-Aguirregomezcorta, J.; Cronin, S.; Donnellan, E.; Fair, S. Progesterone induces the release of bull spermatozoa from oviductal epithelial cells. Reprod. Fertil. Dev. 2019, 31, 1463–1472, doi:10.1071/RD18316.
Ramal-Sanchez, M.; Bernabo, N.; Tsikis, G.; Blache, M.C.; Labas, V.; Druart, X.; Mermillod, P.; Saint-Dizier, M. Progesterone induces sperm release from oviductal epithelial cells by modifying sperm proteomics, lipidomics and membrane fluidity. Mol. Cell. Endocrinol. 2020, 504, 110723, doi:10.1016/j.mce.2020.110723.
In material and methods, it says that 291 oocytes were used in total, but was it per replicate or in all 8 experiments? In the latter case, it would have been 36 oocytes per experiment, and that number is very low when working with porcine IVF.
Reply: Thank you for the comment. The number of oocytes was a priori calculated with a specific software, G*Power 3.1.9.7, to get a final power of our analysis >= 95%. Thus, we set the parameters as follows:
- Effect size: 03
- Probability of α error: 0.05
- Power (1- β error probability: 0.95)
- Number of groups: 8
As output parameters we obtained:
- Noncentrality parameter l: 23.04
- Critical F: 2.0466
- Numerator df: 7
- Denominator df: 248
- Total sample size: 256
- Actual power: 0.9554
Consequently, the calculations indicated that the actual power of 95% will be reached by using 256 oocytes (8 groups of 32 oocytes). Here, for technical reasons, we used a number of oocytes exceeding the requested value of 256, thus we could estimate the actual power of our analysis will be higher than 0.95, which is our target.
Figures 4 and 5 are not adequate to visualize the results or the differences between groups. They do not allow certain calculations to be made, such as the average number of sperm, number of oocytes per replicate...
Reply: We agree. Accordingly, the reviewers’ suggestion we changed the graph typology and the statistical analysis. In particular the layout was defined in keeping with the indication of R1. We added the graphical notation for statistically significative differences.
Line from 302 to 319: This paragraph is not a discussion of your results, it is a protein review (actine/tubuline).
Reply: Thanks for your comment. We have modified the Discussion section, L304-315.
In line 411 it is the first time that the use of estrogens is mentioned. The authors show results of estrogens on POEC, but do not explain how they do the experiment in material and methods.
Reply: We apologize for this mistake. We have modified the Materials and Methods section, L108
The conclusion is not clear if progesterone is good for POEC, or if POEC treated with P4 are better for preparing sperm, or that only p4 on POEC culture does not affect POEC but does affect capacitation. (Already known fact). In any case, there is not a single study on sperm capacitation (motility, protein phosphorylation...).
Reply: We have modified the Conclusion section to satisfy the requirements, and we consider it is clearer now. This study was conducted after the results obtained in previous works from our group and others (Ramal-Sanchez et al., Mol. Cell Endo 2020; Romero-Aguirregomezcorta et al.,Rep. Fert. Develop 2019; Lamly et al., Reproduction 2017), and were mainly focused on elucidating the potential modifications of P4 addition directly to the cells (SOECs). For this reason, and since we agree with this R in the high quantity and quality of the published works regarding sperm capacitation, we did not perform biochemical or other experiments to asses sperm capacitation analysis (motility was always checked and only samples with at least 80% motility were considered for further experiments).
Reviewer 3 Report
The manuscript “Pre-treatment of swine oviductal epithelial cells with progesterone increases the sperm fertilizing ability in an IVF model” was carefully revised. The manuscript evaluates the role of Progesterone supplementation in fertilization media as well as the effect of P4 supplementation in porcine oviduct epithelial cells and its effects during IVF in pigs. The authors concluded that the pretreatment of OECs with P4 ameliorate IVF efficiency in pigs. The field of study and experimental designs are really interesting.
I have a few suggestions to make the manuscript more reader-friendly.
- The graphs presented in figures 4 and 5 are not easy to follow, I suggest the authors to use traditional bar graphs showing each group. In addition, I suggest the authors to stratify the IVF parameters in “Not fertilized”, “Polyspermic” and “Normal fertilized”. Together, these three parameters should represent 100% of the oocytes submitted to fertilization and would help to give a better general idea of the results.
- It would also be interesting to see the mRNA expression levels of more regulators of IVF efficiency in the OECs cells cultured with P4 such as Ca+ channels regulators and secondary messengers of cell signaling to try to explain the regulation and mechanisms that correlate with the findings.
- How the authors explain the downregulation of the three evaluated genes in OECs after P4 treatment? Wouldn’t be expected to observe an upregulation of SSP1 and DMBT1? Are the authors sure that this could not be a reflex of the reference gene used in the analysis?
- Are there statistical differences in the data presented in figure 6? If yes, please modify graphs accordingly.
Author Response
The Authors thank the Editor and Referees for their careful and rapid review of our manuscript. All the modifications suggested by the Referees have been appreciated and followed, and we consider that the manuscript has considerably improved now. Please find enclosed a clean version of the revised manuscript (Animals_cleanversion) and a file with all the highlighted modifications (Animals_highlighted). As requested by the Referees, the figures regarding the IVF results have been modified and better explained. The Conclusion section has been extensively revised.
We really hope that you could reconsider our new version of the manuscript suitable for publication in Animals. We remain at your disposal in case you need any other clarification or additional information.
Looking forward to hearing from you, we send you our best regards.
Reviewer#3: The manuscript “Pre-treatment of swine oviductal epithelial cells with progesterone increases the sperm fertilizing ability in an IVF model” was carefully revised. The manuscript evaluates the role of Progesterone supplementation in fertilization media as well as the effect of P4 supplementation in porcine oviduct epithelial cells and its effects during IVF in pigs. The authors concluded that the pretreatment of OECs with P4 ameliorate IVF efficiency in pigs. The field of study and experimental designs are really interesting.
I have a few suggestions to make the manuscript more reader-friendly.
The graphs presented in figures 4 and 5 are not easy to follow, I suggest the authors to use traditional bar graphs showing each group. In addition, I suggest the authors to stratify the IVF parameters in “Not fertilized”, “Polyspermic” and “Normal fertilized”. Together, these three parameters should represent 100% of the oocytes submitted to fertilization and would help to give a better general idea of the results.
Reply: We thank Reviewer 3 for her/his careful reading and smart comments and suggestions, which we agree with and that were taken into consideration. According to the Reviewers’ suggestions, we changed the graph typology and the statistical analysis. In particular the layout was defined in keeping with the indication of R1. We added the graphical notation for statistically significative differences.
It would also be interesting to see the mRNA expression levels of more regulators of IVF efficiency in the OECs cells cultured with P4 such as Ca+ channels regulators and secondary messengers of cell signaling to try to explain the regulation and mechanisms that correlate with the findings.
Reply: Even if this was out of the scope of the present work, we agree with R3 and we really appreciate this suggestion, that we would like to study in a more detail in a further work.
How the authors explain the downregulation of the three evaluated genes in OECs after P4 treatment? Wouldn’t be expected to observe an upregulation of SSP1 and DMBT1? Are the authors sure that this could not be a reflex of the reference gene used in the analysis?
Reply: Previous studies have shown that an increased expression of some oviductal proteins, such as oviductin (OVGP1), osteopontin (SPP1) or deleted in malignant brain tumors 1 (DMBT1), reduces the rate of polyspermy in the pig, through different mechanisms, thus resulting in an increase of normal fertilization in vitro (see Coy & Yanagimachi, 2015).
In this study we have shown that the administration of progesterone (P4) (100ng/ml) increases the sperm fertilizing ability due to the higher percentage of polyspermic oocytes (fig. 5).
In this regard, the increased incidence of polyspermy observed in our IVF swine model is fully concordant with the slight decrease in mRNA expression of OVGP1, SPP1 and DMBT1 (fig.6), following the P4 (100ng/ml) administration.
In concordance with these results, a previous study has demonstrated that increased levels of serum progesterone induce an evident inhibitory effect on OVGP1 mRNA expression, in oviductal mucosal tissue from fertile women (Regulation of human oviductin mRNA expression in vivo. C Briton-Jones, I H Lok, P M Yuen, T T Chiu, L P Cheung, C Haines. Fertil Steril. 2001 May;75(5):942-6. doi: 10.1016/s0015-0282(01)01696-x).
Nevertheless, as mentioned in the text, we are aware that protein analysis is needed to confirm the gene expression results.
Are there statistical differences in the data presented in figure 6? If yes, please modify graphs accordingly.
Reply: In keeping with the Reviewers’ claim, we have changed the graphs layout and the statistical treatment of the data. In detail, the data analysis was realized by adopting a non-parametric approach, since the results of D’Agostino and Pearson Normality test indicated a non-gaussian distribution of the data and the controls are characterized by a variance = 0 (by definition the CRTL value is 1). Thus, we used the Kruskal Wallis test, whose results are reported in the new graphs, where the data are expressed as median +/- range.
Round 2
Reviewer 1 Report
The revision of the manuscript made by the authors has notably improved the final results. Authors have changed figure 3 and the graphics in figure 4 and 5.
Nevertheless, I think the figure 4 must be improved and probably the statistical analysis should be checked. In figure 4 results are shown as the fall increase respect to control samples. It is hard to beliebe that with those minimum standards errors authors did not find any statistical differences in P4 10 and 1000ng/mL with respect to control when polyspermic sperm or nº sperm/polyspermic oocyte were analyzed. I suggest to authors including the real values of the experiments (actually it gives more information to the readers) and/or remake the statistic.
A second view of the figure 5 is telling us that the treatment with p4 doesn´t have any effect on sperm capacitation process since not differences were found with respect to control. Nevertheless, in figure 4 clear differences were shown. How authors can explain these opposite results?
A suggest to include colours to apreciate much better the graphs.
Author Response
Reviewer #1: The revision of the manuscript made by the authors has notably improved the final results. Authors have changed figure 3 and the graphics in figure 4 and 5. Nevertheless, I think the figure 4 must be improved and probably the statistical analysis should be checked. In figure 4 results are shown as the fall increase respect to control samples. It is hard to believe that with those minimum standards errors authors did not find any statistical differences in P4 10 and 1000ng/mL with respect to control when polyspermic sperm or nº sperm/polyspermic oocyte were analyzed. I suggest to authors including the real values of the experiments (actually it gives more information to the readers) and/or remake the statistic.
Reply: We thank Reviewer 1 for her/his careful reading and comments and suggestions. Accordingly, we have added the statistical significance within the graph. We would like to point out that the histograms (Fig. 4 and 5) show the standard deviation and not the standard error. Data from Figure 4 were analyzed using the Turkey's multiple comparison test, finding statistical differences among the groups (p<0.01, highlighted in the graph). In this case, data are expressed as the delta value of the difference between the CTRL and treated samples.
To assess the effect of different treatments on IVF we carried out four independent experiments using samples from different animals (biological and technical replicates).
A second view of the figure 5 is telling us that the treatment with p4 doesn´t have any effect on sperm capacitation process since not differences were found with respect to control. Nevertheless, in figure 4 clear differences were shown. How can authors explain these opposite results? A suggest to include colors to appreciate much better the graphs.
Reply: Thank you for your comment. While Figure 4 shows the value of delta (i.e., the difference between the CTRL and the treated samples), Figure 5 shows the comparison among the group., evidencing that the best performance was obtained when using P4 at 100 ng/mL due to the greater values obtained in the percentage of polyspermic oocytes. The histograms shows that there is a significant difference (Dunnett’s multiple comparison test) between the control group and the group with SOECs treated with p4 100 ng/mL and without the addition of P4 during capacitation. The significance was now included in the graph; we apologize for this mistake.
Regarding the choice of including colors or not, we appreciate very much your suggestion. However, we decided to keep the grey-scale as a form of inclusiveness, mostly for color-blind readers.
Reviewer 2 Report
The authors have answered my questions and corrected the errors.
Author Response
Thanks for the comment
Reviewer 3 Report
1. I don’t understand figure 4 and I can not see the beneficial effects of the 100ng/ml of P4 supplementation to IVF media regarding IVF outcomes. Why control group is not present as one of the results (one of the bars) in the graph the same way as figure 5?
2. Figure 5 clearly shows that the P4 supplementation in OECs cell culture has no beneficial effect at all to the IVF outcomes.
Author Response
Reviewer #3: I don’t understand figure 4 and I can not see the beneficial effects of the 100ng/ml of P4 supplementation to IVF media regarding IVF outcomes. Why control group is not present as one of the results (one of the bars) in the graph the same way as figure 5?
Reply: Thank you very much for your valuable comments and suggestions. Figure 4 illustrates the dose-response relationship between SOECs treated with P4 at different concentrations and IVF outcome, expressed as ΔIVF (calculated as the difference between the data referred to CTRL and treated samples), for this reason the control group is not present in the graph. On the contrary, Figure 5 shows all the groups, including the CTRL group, and the results are expressed in percentage.
Figure 5 clearly shows that the P4 supplementation in OECs cell culture has no beneficial effect at all to the IVF outcomes.
Replay: Thank you for your comment. In fact, we apologize for this mistake, we have now included in a clearer way the significance within the graph. Here, we used the IVF as a functional test for spermatozoa, without the aim of obtaining embryos. It means that our work was set up to evaluate the capacitation status of sperm cells in a specific context: after the SOECs treatment with P4. As it has been reported in literature, swine in vitro maturated oocytes are partially unable to avoid the polyspermic fertilization. In these specific conditions, it is possible to consider the number of polyspermic oocytes as a marker of the fertilizing ability, considering the ratio sperm/oocytes and the length of coincubation. Figure 5 shows that the best performance was obtained when using P4 at 100 ng/mL due to the greater values obtained in the percentage of polyspermic oocytes.
Round 3
Reviewer 3 Report
My comments on the last review were that I do not agree with authors conclusions that progesterone ameliorate IVF outcomes in porcine IVF. Their figures (4 and 5) clearly show that progesterone has no effect at all on IVF. In addition figure 4 is confusing, since CTR group is not really being compared to other groups. Nevertheless, no change on that subject, has been done in this last version of the manuscript that you have sent me by email. For that reason, on my end, I suggest for the rejection of the manuscript accordingly my last review.
Author Response
Reviewer 3
My comments on the last review were that I do not agree with authors conclusions that progesterone ameliorate IVF outcomes in porcine IVF. Their figures (4 and 5) clearly show that progesterone has no effect at all on IVF. In addition figure 4 is confusing, since CTR group is not really being compared to other groups. Nevertheless, no change on that subject, has been done in this last version of the manuscript that you have sent me by email. For that reason, on my end, I suggest for the rejection of the manuscript accordingly my last review.
Reply: We thank R3 one more time for the time implied in reviewing our manuscript. We have revised the whole manuscript to edit the English grammar and style. As we tried to explain in our previous “Response to Reviewers”, in this work we used the IVF as a functional test for spermatozoa. It means that our work was set up to evaluate the capacitation status of sperm cells in a specific context: after the SOECs treatment with P4. As it has been reported in literature, swine in vitro maturated oocytes are partially unable to avoid the polyspermyc fertilization, thus in specific conditions and in terms of sperm/oocytes ratio and length of coincubation, it is possible to obtain polyspermic oocytes as a marker of their fertilizing ability. With this information, we found that the best performance, due to the greater values obtained in the percentage of polyspermic oocytes and number of spermatozoa/polyspermic oocytes that allow to infer the sperm fertilizing ability, was obtained when using P4 at 100 ng/mL.
As much as is regard the Figure 4, it graphically represents the effects of P4 at different concentrations in three main parameters: the fertilization rate, the number of polyspermic oocytes and the number of spermatozoa per polyspermic oocyte. The differences found are expressed in percentage, calculated as the delta value. The delta value calculates the overall change of a value, in our case comparing the control (CTRL) with the three parameters studied (fertilization rate, number of polyspermic oocytes, number of spermatozoa per polyspermic oocyte). Thus, CTRL does is compared with the others groups, otherwise delta value could not had been calculated. The groups were then analyzed using a Turkey’s test, as an additional information to show the statistical significance among the groups of study (not with the control, because they are already compared to the control).
On the other hand, Figure 5 was analyzed following a different strategy, since it is a different set of experiments with different conditions. In this case, delta value was not calculated, but the “classical” comparison was performed. Groups were then compared with a Dunnett test, finding a statistically significative difference when spermatozoa were capacitated without P4, on SOECs pre-treated with P4.
The results gathered here does not try to prove or show that “progesterone ameliorates IVF outcomes in porcine IVF”. Our results show that there is an important functional link between sperm capacitation and two major modulators of the sperm physiology, as such are SOECs and P4. Moreover, and despite the limitations, we have showed that the incubation of male gametes with SOECs pre-treated with P4 at a certain concentration is able to increase their fertilizing ability, standing up the need for further studies about the mechanism of action.
We hope this response could be considered satisfactory and that R3 could agree with us now. We apologize for the inconveniences if we were not so clear in the previous responses.
